# Protected 911: Development, Implementation, and Evaluation of a Prehospital COVID-19 High-Risk Response Team

**DOI:** 10.3390/ijerph19053004

**Published:** 2022-03-04

**Authors:** Justin Mausz, Nicholas A. Jackson, Corey Lapalme, Dan Piquette, Dave Wakely, Sheldon Cheskes

**Affiliations:** 1Peel Regional Paramedic Services, 1600 Bovaird Drive East, Brampton, ON L6V 4R5, Canada; nicholas.jackson@peelregion.ca (N.A.J.); corey.lapalme@peelregion.ca (C.L.); dan.piquette@peelregion.ca (D.P.); dave.wakely@peelregion.ca (D.W.); 2Department of Health Research Methods, Evidence, and Impact, McMaster University, 1280 Main Street West Room HSC-2C1, Hamilton, ON L8S 4K1, Canada; 3Sunnybrook Centre for Prehospital Medicine, 77 Brown’s Line, Suite 100, Toronto, ON M8W 3S2, Canada; sheldon.cheskes@sunnybrook.ca; 4Division of Emergency Medicine, University of Toronto, 6 Queen’s Park Cres. W., Toronto, ON M5S 3H2, Canada

**Keywords:** paramedic, COVID-19, patient safety, personal protective equipment, occupational health and safety, quality improvement

## Abstract

Patients with COVID-19 who require aerosol-generating medical procedures (such as endotracheal intubation) are challenging for paramedic services. Although potentially lifesaving for patients, aerosolizing procedures carry an increased risk of infection for paramedics, owing to the resource limitations and complexities of the pre-hospital setting. In this paper, we describe the development, implementation, and evaluation of a novel pre-hospital COVID-19 High-Risk Response Team (HRRT) in Peel Region in Ontario, Canada. The mandate of the HRRT was to attend calls for patients likely to require aerosolizing procedures, with the twofold goal of mitigating against COVID-19 infections in the service while continuing to provide skilled resuscitative care to patients. Modelled after in-hospital ‘protected code blue’ teams, operationalizing the HRRT required several significant changes to standard paramedic practice, including the use of a three-person crew configuration, dedicated safety officer, call–response checklists, multiple redundant safety procedures, and enhanced personal protective equipment. Less than three weeks after the mandate was given, the HRRT was operational for a 12-week period during the first wave of COVID-19 in Ontario. HRRT members attended ~70% of calls requiring high risk procedures and were associated with improved quality of care indicators. No paramedics in the service contracted COVID-19 during the program.

## 1. Introduction

In the early months of the coronavirus disease 2019 (COVID-19) pandemic, uncertainty surrounding the clinical course of patients and, importantly, the transmissibility of the pathogen presented significant operational challenges for paramedic services. Initial reports suggested that more than 40% of hospitalized patients would require supplemental oxygen and as many as 15% may require mechanical ventilation [1,2,3]—procedures for which there is an established risk of disease transmission to healthcare providers through infectious aerosols [4]. 

Indeed, the initial weeks of the pandemic saw shocking rates of infections and deaths among healthcare workers caring for patients with COVID-19 [5,6]. This risk of infection was potentially even more pronounced for paramedics, owing to the complexities of providing care in a prehospital setting, where environments are difficult to control, patients are undifferentiated, resources are limited, and logistical issues (such as inclement weather or confined spaces) can make wearing Personal Protective Equipment (PPE) challenging. Concern for paramedic safety from communicable disease is not without precedent [7]. 

In the aftermath of the 2002–2003 Severe Acute Respiratory Syndrome (SARS) outbreak in Toronto, one investigation found that healthcare providers who contracted SARS were more likely to be paramedics [8]. The SARS outbreak also resulted in large portions of the paramedic workforce quarantined at home after high-risk patient exposures [9]—a necessary safety precaution that nevertheless placed a significant strain on service delivery. The concern in our service, then, was the potential to encounter large numbers of patients with COVID-19 who would require Aerosol-Generating Medical Procedures (AGMPs), placing paramedic safety at risk on an individual level, and straining resources on a system level.

While AGMPs carry an increased risk of infection for healthcare providers, the procedures can be lifesaving for patients. This presented a dilemma for our paramedic service in terms of balancing the safety of our personnel with our responsibility to care for critically ill patients, many of whom were anticipated to require AGMPs. This challenge was certainly not limited to our own service as jurisdictions around the world struggled with similar ethical and legal dilemmas in light of dramatic imbalances between system demand and available resources [1]. 

In an effort to optimize both provider and patient safety, our paramedic service developed a novel specialized response team with the mandate of providing care for patients suspected to have COVID-19 who required aerosolizing procedures. Following the Standards for Quality Improvement Reporting Excellence (SQUIRE) reporting guidelines [10], we describe the development, implementation, and evaluation of a COVID-19 High Risk Response Team (HRRT) in Peel Region, Ontario, Canada. Although not specifically intended as an interventional trial, our goal is to provide a rich description of the program and share lessons we learned in its development and implementation.

## 2. Methods

### 2.1. Context

Peel Regional Paramedic Services (PRPS) is the publicly funded, sole provider of land ambulance and paramedic services to the municipalities of Brampton, Mississauga, and Caledon, Ontario, collectively encompassing a mixed suburban and rural geography of 1200 km^2^, with a population of 1.5 million residents. The service employs more than 700 Primary and Advanced Care Paramedics (PCP/ACP) who staff a total of 65 ambulances and eight rapid response units during peak hours. 

The Region of Peel is geographically divided into four quadrants, each with a centralized headquarters facility from which paramedic crews start their shift and are then sent to smaller community stations in a ‘hub-and-spoke’ system of deployment. Ambulances are normally crewed by two paramedics in a PCP-PCP or PCP-ACP configuration and rapid response units are crewed by a single PCP. On average, PRPS responds to approximately 130,000 emergency calls per year, of which typically 1200 are for patients in cardiac arrest.

Peel Region has persisted as one of several ‘hot spots’ in the province for community transmission of COVID-19 [11,12], owing, in part, to large populations of essential workers and intergenerational households [13]. At this early point in the pandemic, vaccine development and distribution in Ontario were still several months away. In April of 2020, operating within a declared state of emergency in the province, PRPS mobilized a joint union and management Task Force [14] under the incident management system framework to develop the HRRT. Our task force included representatives with expertise in paramedic education (C.L., J.M.), occupational health and safety (D.P.), service operations and leadership (N.A.J.), with D.W. serving as the liaison with the Emergency Operations Centre (EOC) charged with coordinating the service’s response to the crisis.

### 2.2. Interventions

#### 2.2.1. Program Goals and Objectives

The overall program goal for the HRRT was to prevent work-related COVID-19 infections among paramedics in our service while continuing to provide high-quality resuscitative care to patients. In support of that goal, we identified three specific program objectives:(1)Limit paramedic exposure to AGMPs within the service to the members of the HRRT to the greatest extent reasonably possible.(2)Ensure the safety of the members of the HRRT given (what we anticipated would be) their concentrated and frequent exposure to high-risk patient types and procedures.(3)Provide high quality resuscitative care to critically ill patients, including the performance of aerosolizing procedures where they were typically indicated.

Early in the course of the pandemic, the regulatory bodies that govern paramedic practice in Ontario released a series of memos that recommended curtailing specific AGMPs in all but extraordinary circumstances [15]. This included limiting the use of nebulized medications, continuous positive airway pressure, bag-mask ventilation, and high flow oxygen administration, among others.

#### 2.2.2. Operational Concept for the HRRT

In defining a conceptual basis for the team’s operations, we drew on the procedures for ‘Protected Code Blue’ teams [16,17,18] that were increasingly being used in-hospital in response to the pandemic. We also conducted a non-systematic narrative review of the emerging literature describing recommendations for airway management and cardiopulmonary resuscitation in patients confirmed or suspected to have COVID-19 [19,20], distilling the literature into a series of recommendations that informed our operations (Table 1). 

In considering these recommendations, we were able to structure the team to achieve most of them, with the notable exception of using video laryngoscopes for intubation. Video laryngoscopy is not standard in our service, and the time pressure to launch the team would have made introducing the new skill challenging. Finally, we drew on the best practices related to human factors [21,22] used by high-performing emergency medical services systems in Europe and Australia, such as the use of pre-procedure checklists and principles of crisis resource management. 

Our task force developed the operational concept and procedures for the team through consensus building based on the emerging literature and best practice recommendations described above. This required regular in-person meetings, frequent engagement with relevant stakeholders (e.g., front line paramedics and supervisors from other departments) within the paramedic service, and simulating proposed procedures to identify potential failure points.

### 2.3. Team Selection and Training

The philosophy underpinning the operations for the team was high performance, safety-oriented teamwork. Operationalizing this required several significant changes to the paramedic practice in our context, and we wanted to recruit paramedics who not only had strong clinical skills but also had experience working in small teams. 

Our development task force circulated a call for applications and ranked candidates based on their professional experience in four domains of interest: clinical excellence (as demonstrated, for example, through teaching responsibilities or continuing professional development), teamwork, leadership, and health and safety experience. Candidates were additionally required to have a minimum of two years of experience in their respective provider classification (e.g., primary or advanced care).

Once our final cohort was selected, the team members (n = 54) completed a combination of online and in-person (didactic and hands-on) onboarding training totaling 8 h of curricular time. The training focused on the pathophysiology of COVID-19, principles of patient management, and the foundations of high-performance teamwork. We adapted our patient care procedures to de-emphasize bag-mask ventilation in favor of an escalating oxygen therapy regimen [23]. 

Where patients failed escalating oxygen therapy, our management strategy favored early endotracheal intubation to facilitate definitive control of the airway. Later, as consensus guidelines continued to evolve [24], we adapted our practices to reintroduce continuous positive airway pressure (CPAP) after consultation with online medical control for awake patients who had not responded to escalating oxygen therapy. Finally, we trained HRRT members in the use of mechanical cardiopulmonary resuscitation (CPR) as part of cardiac arrest resuscitation. 

Although not standard in our service, this adaptation was in line with interim recommendations from the American Heart Association (AHA) [25] as a means of reducing the number of providers required at a resuscitation. During cardiac arrests, we advocated for continuous chest compressions with a filtered non-rebreather oxygen mask and oropharyngeal airway in place until an endotracheal tube or (as a rescue) laryngeal mask airway was inserted.

### 2.4. Safety Procedures

To maximize both patient and provider safety, we undertook a comprehensive review of the steps involved in each ‘high-risk’ aerosolizing procedure (Table 2) the team members would be performing and developed a series of pre-procedure call–response checklists. The checklists (available online as Appendix A; see Figure 1 for an example) were developed through a review of expert/consensus guidelines and printed on waterproof paper and in large font with contrasting colors and images for use on calls.

In contrast to normal operations where ambulances in our service are crewed by two paramedics of either provider classification, we defined an optimal HRRT crew configuration as being two ACPs and one PCP Safety Officer. The two ACPs were responsible for providing direct patient care, while the safety officer was intended to coordinate scene logistics and oversee the safety of the team. 

Safety officers were removed from patient care and trained to maintain line-of-sight with the treating paramedics at ~2 m from the patient. This gave the safety officer a ‘big picture’ view of the scene and allowed them to monitor the encounter for safety concerns, such as a breach in PPE. When the treating paramedics were ready to perform a high-risk AGMP, the safety officer would guide the crew through the specific call–response checklist. Once marked ‘checks complete’, the treating paramedics were clear to perform the procedure.

Our HRRT members adopted a higher level of PPE than was commonly being used by our paramedics due to their concentrated exposure to high-risk patient types. This included an inner layer of gloves and a full-body Tyvek suit with a hood, covered by an outer fluid-impermeable gown and a second pair of extended cuff gloves. For respiratory protection, HRRT crews wore Powered Air-Purifying Respirators (PAPRs) with a reusable helmet that completely covered the paramedic’s face (Figure 2). 

The selection of respiratory protection aligned with early guidelines for airway management that recommended the use of an N95 (or FFP3) respirator. Several sources [18,19,26,27,28] recommended PAPRs specifically. The use of reusable respiratory protection for our team members also allowed us to conserve the disposable respirator supply within the service in what was, at the time, a period of scarcity. HRRT members used checklists to both don and doff PPE and conducted a ‘buddy check’ prior to entering a scene.

### 2.5. Operations

We designed the HRRT to deploy one three-person crew per geographic quadrant of the region 24 h per day. In a departure from normal operations, HRRT crews were utilized outside of regular dispatching procedures and were autonomous in selecting calls to attend. This required HRRT crews to be mobile and monitoring emergency calls that were occurring in the system to identify high acuity patients that were likely to require aerosolizing procedures. The crews received text message alerts from our dispatch center for calls involving ‘obvious immediate threat to life’ criteria, including cardiac arrest, altered level of consciousness, and severe respiratory distress, among others. Other paramedics in the system could additionally request an HRRT crew to attend a scene.

The HRRT was generally intended to be a promptly arriving ‘second-in’ resource. When an HRRT crew arrived on scene, the crew would conduct a risk assessment with the paramedics already present and, where the patient was anticipated to require AGMPs, assume care of the patient and control of the scene. We adapted ‘Haz-Mat’ procedures in designating a ‘Hot Zone’, in which only HRRT personnel were present during patient care, and a ‘Warm Zone’ where other responders in airborne PPE would ‘stage’ to help with scene logistics. 

Once any aerosolizing procedures had been completed, other responders at the scene would assist with extrication, and the patient would be transported to hospital. At the hospital, the safety officer was responsible for liaising with the receiving facility staff to determine a transfer of care plan before the patient was removed from the vehicle. Following each mission, HRRT crews conducted an informal, semi-structured ‘after action review’ using a series of prompts that focused on enhancing crew safety, team efficiency, and the quality of patient care.

### 2.6. Study of the Interventions

Our approach to program evaluation involved blueprinting specific data sources and performance metrics to assess each of our three objectives (see Appendix A) and evaluating performance on a weekly basis using the findings to inform ongoing changes to our operations.

## 3. Measures

Our program evaluation consisted of a manual review of all electronic patient care records (ePCRs) completed by our personnel as well as the ePCRs of ‘missed cases’ where an HRRT crew was unavailable to attend a call where high-risk AGMPs were required. We were specifically interested in the proportion of calls involving high-risk AGMPs that our team members attended (Objective 1); documented safety breaches and lost days from work as a result of quarantine or illness (Objective 2); and procedural success rates for airway maneuvers as an indicator of quality of care (Objective 3). 

To provide a contextualized basis of comparison, we also manually reviewed the ePCRs for all calls in which high-risk AGMPs were performed in the first six weeks of the pandemic leading up to the launch of the team. Following the conclusion of the program, we surveyed our personnel to explore questions related to mental health, perception of safety, and job satisfaction as a result of working on the team.

## 4. Analysis

We used summary and descriptive statistics to characterize our data, using Analysis of Variance (ANOVA), Pearson correlation, and Chi-Square tests to evaluate individual components of each program objective, with confidence intervals (where applicable) set at 95%. Specifically, for Objective 1, we explored whether our high-risk AGMP capture rate and related performance metrics (such as response time) were affected by staffing. For Objective 2, we compared administrative data to evaluate both the likelihood of being placed on quarantine and the mean number of lost shifts or regular duty versus HRRT paramedics. For Objective 3, we compared utilization and success rates for advanced airway maneuvers for before and after the launch of the team and in captured versus missed cases. 

We were interested in exploring whether patients attended by HRRT crews were more likely to have an advanced airway attempted when indicated and whether the attempt was more likely to be successful. Medical directives in our region governing advanced airway maneuvers define an ‘attempt’ as the insertion of a laryngoscope or supraglottic airway into the mouth and limit the number of attempts to two. Finally, for qualitative data in the form of free-text survey comments, we used qualitative content analysis [29] to thematically code segments of text. We followed convention in setting a *p*-value threshold of 5% for statistical significance; however, as this was not intended as an interventional trial, we did not conduct an a priori power calculation.

## 5. Ethical Considerations

This program evaluation was exempted from the requirement for a research ethics board review by the Sunnybrook Research Institute under Article 2.5 (concerning quality assurance, improvement, or program evaluation activities) of the Tri-Council Policy Statement for the ethical conduct for research involving humans (TCPS-2).

## 6. Results

The HRRT was launched 18 days after the mandate to develop the team was given. This included the conception and design of the program, recruitment and training of personnel, development of the safety and operational procedures, procurement of equipment, and communication to stakeholders. The program was operational for a 12-week period from 21 April through 12 July 2020, with the decision to stand the team down coinciding with the declining case counts toward the end of the first wave in the province of Ontario.

### 6.1. Operations

HRRT crews attended a total of 1244 calls potentially meeting the mandate of the team based on dispatch information, of which 735 patients (59%) did not require AGMPs, leaving a final sample of 509 patients for which the HRRT crew assumed care. This corresponded to an overall proportion of 2.8% of calls occurring in the system during the 12-week program.

Of the 509 patients, 51% were categorized as Canadian Triage and Acuity Scale [30] (CTAS) level 1 (requiring resuscitation), and 35% were classified as CTAS level 2 (‘emergent’). HRRT crews performed aerosolizing procedures in 60% of patient encounters and high-risk procedures in 45% of patients, arriving on scene first or simultaneously with the first-dispatched ambulance in 150 (31%) of cases with an average response time of 7.83 (95% Confidence Interval [CI] 7.46–8.20) min. 

When not first on scene, HRRT crews arrived within an average of 5.22 (95% CI 4.61–5.82) minutes of the first-arriving ambulance. HRRT crews transported 61% of patients, with 24% being pronounced dead at the scene, and the remainder (~15%) either refusing care (n = 9) or being transported in another ambulance (n = 59). This often occurred, for example, when a patient had already been transported by the first-arriving paramedics and HRRT members provided care en route to hospital.

#### 6.1.1. Objective 1: Limit Paramedic Exposure to AGMPs

Over the 12-week program, there were a total of 290 calls that involved *high-risk* AGMPs, with HRRT attending 198 for an overall capture rate of 68.2%. This proportion of patients requiring high-risk AGMPs accounted for 1.1% of the total calls occurring in the system.

The HRRT capture rate varied by week ranging from 39% (week 11) to 86% (week 8) with an average weekly capture rate of 67.1% (SD 13.38) and a missed case rate of approximately one case per 24-h period. When evaluating the potential impact of shift staffing on caseload, we observed a relationship between the total staffing hours and the number of calls per shift (*r* = 0.57, *p* = 0.01), suggesting that, as staffing hours increased, the number of calls the HRRT crews attended increased proportionally. 

However, we did not observe a relationship between staffing hours and the number of missed cases involving high-risk AGMPs, either correlationally as a function of the total staffing hours (*r* = −0.04, *p* = 0.591) or when comparing missed cases on optimally vs. not-optimally staffed shifts (mean 0.65 (95% CI 0.48–0.82) vs. 0.44 (95% CI 0.30–0.58), *p* = 0.069). Additional performance indicators stratified by program week are available online as Appendix A.

#### 6.1.2. Objective 2: Team Safety

There were two documented breaches in PPE during the program. The first occurred when part of a member’s fluid-impermeable gown was drawn into the air intake on their PAPR causing the respirator to fail during patient care. The second occurred when a member slipped and collided with a wall while carrying a patient down a flight of stairs, breaking the cover on their PAPR in the process. In both cases, the members were placed on 14-day quarantine and neither developed COVID-19. We also subsequently switched gowns from a plastic to a cloth model to reduce the likelihood of the material causing a PAPR failure.

No paramedics in the service developed COVID-19 during the program. Lost time due to quarantine among HRRT members was generally comparable to regular duty paramedics in the system. In total, 10 (out of 43 active; 23%) team members were placed on quarantine during the program compared with 127 (out of 592; 21%) regular duty paramedics (OR 1.10, 95% CI 0.53–2.31, *p* = 0.07), losing an average of 5.40 (Standard Deviation [SD] 2.27) and 5.41 (SD 4.56) shifts, respectively (*p* = 0.995). Administrative data, however, did not specify whether the reason for quarantine was attributable to a work-related exposure.

#### 6.1.3. Objective 3: Procedural Success Rates

In the 6 weeks between the declaration of the pandemic by the World Health Organization on 11 March 2020 and the launch of the team on 21 April 2020, advanced airway maneuvers (endotracheal intubation or supraglottic airway insertion) were attempted in 51% of indicated (most commonly cardiac arrest) cases with an overall documented success rate of 89%. Endotracheal intubation was the method selected in 58% of cases, with an overall documented success rate of 67%. This proportion of cases in which an advanced airway was attempted when indicated trended downward from 67% in pre-HRRT week 1 to 53% in pre-HRRT week 6, with the lowest proportion (37%) in pre-HRRT week 3.

Following the launch of the team, the proportion of cases in which advanced airway maneuvers were attempted when indicated increased to 94%, varying slightly by week, however, with an average of 93% and an overall success rate of 98%. Endotracheal intubation was the most commonly selected method, used in 79% of cases with an overall documented success rate of 81%. When evaluated categorically, following the launch of the team, advanced airway maneuvers were more likely to be attempted (OR 13.19, 95% CI 6.39–27.24, *p* < 0.001) and successful (OR 6.43, OR 1.54–26.75, *p* = 0.008); however, the selection of endotracheal intubation (OR 1.88 95% CI 0.90–3.93, *p* = 0.90) and successful intubation (OR 2.07, 95% CI 0.90–4.76, *p* = 0.08) did not reach significance at the 5% threshold.

Compared with missed cases, calls in which the HRRT attended were more likely to have an advanced airway attempted (41% vs. 93%, OR 19.76, 95% CI 8.91–43.85, *p* < 0.001), the attempt was more likely to be successful (68% vs. 98%, OR 23.21, 95% CI 5.61–95.91, *p* < 0.001), endotracheal intubation was more likely to be selected (47% vs. 79%, OR 4.24, 95% CI 1.51–11.91, *p* = 0.007), and attempts at endotracheal intubation were more likely to be successful (47% vs. 81%, OR 4.87, 95% CI 1.72–13.79, *p* = 0.003). Additional performance indicators (Appendix A) stratified by program week are available online as Appendix A.

#### 6.1.4. Team Member Experiences: Survey Comments

The novelty of the program combined with its speed of deployment made its introduction into the system challenging for several reasons. A three-person crew configuration, self-dispatching, new procedures (i.e., call–response checklists), and equipment (i.e., mechanical CPR and enhanced PPE) were all brought online in a time of considerable uncertainty amid significant societal disruption. This was reflected in the survey comments from our members who cited interpersonal conflict with supervisors and colleagues on regular ambulances, who they felt did not understand the role of the team.

Despite the stressors, the survey comments were generally quite positive, particularly on questions related to perceptions of safety, mental health, and job satisfaction. Even though the team’s mandate was to respond to ‘high-risk’ calls, the multiple layers of redundant safety features meant that the team members reported feeling safer working on the team. Many also explained that the expectation of attending only the highest acuity calls prompted them to engage more deliberately in continuing professional development to be “at the top of (their) game”, finding the experience intensely rewarding on a professional level. 

One potential area of concern was the team members’ concentrated exposure to high acuity patients and the risk of critical incident stress [31] given, for example, that more than 85% of their patients were critically ill, and 25% of patients ultimately died in their care. When asked about their mental health, the members reported that working on a small team in which calls were debriefed promptly offered several psychosocial supports that ameliorated the potential impact of this increased exposure to critical illness.

## 7. Discussion

One of the key recommendations to come out of the SARS Commission Report of 2007 was that, in the absence of scientific certainty regarding the risk of infection to healthcare providers, the ‘precautionary principle’ should guide decisions around provider safety [32]. We developed the HRRT early in the pandemic in an effort to optimize both paramedic and patient safety, given the initially limited knowledge about COVID-19 and its risks to healthcare providers. 

Our overarching goal was to prevent work-related infections within the service, and although no paramedics (on or off the team) tested positive during the program because of a patient encounter, the observational nature of the intervention makes drawing causal inferences inappropriate. As the pandemic progressed, stabilization of the PPE supply, universal masking in communal workspaces, comprehensive infection control, prevention guidelines, vaccination, and the establishment of a novel ‘quarantine support unit’ within the service all contributed to an enhanced organizational response to the crisis. 

The result has been that, in the year since the team’s deactivation, there have been no cases of paramedics within our service contracting COVID-19 as a result of a patient encounter. Reports of COVID-19 infection among paramedics in North America are also reassuringly low [33]. The Region of Peel’s corporate finance department estimated the total operating cost of the HRRT program at ~2 million CAD. Staffing costs were the largest expenditure as the team represented an operational enhancement that necessitated back-filling the regular-duty positions the HRRT members would normally occupy. This meant that when considering the resource requirements of the HRRT in terms of operating costs, personnel, and equipment, the cost–benefit calculation did not support reactivation of the team. There are, however, a number of lessons garnered through the development process that are worth sharing.

First, the downward trend that we observed in advanced airway maneuvers in the weeks leading up to the launch of the team is noteworthy. In early March, paramedics attempted advanced airway insertion in approximately 70% of patients in cardiac arrest; however, this proportion decreased significantly over the following weeks, reaching a low of 35% toward the end of March and early April. This trend coincided with increasing case counts and growing media coverage of the pandemic and occurred at a time when provincial regulatory bodies governing paramedic care were specifically recommending early insertion of an advanced airway during resuscitation. 

Although we did not explore why specifically, it is reasonable to infer that concerns over provider safety may have had some role in the declining use of procedures known to be high risk. The survey feedback we received from our team members indicated that HRRT personnel felt more comfortable performing higher risk procedures, given the multiple layers of safety precautions. To our knowledge, the degree to which perceived risk to healthcare providers may influence treatment decisions has not been studied in the context of COVID-19 and warrants investigation.

Second, the HRRT provides an interesting case study in considering the role of specialization in pre-hospital care. Only 1% of calls occurring in the system met the mandate of the team, and even when including all patients classified as CTAS Level 1 (requiring resuscitation), regardless of the need for AGMPs, the proportion (2.2%) of patients with life-threatening illness was small. This parallels a number of studies showing paramedic exposure to high acuity cases is infrequent [34,35,36,37], and we have abundant literature highlighting the challenges of maintaining competence in high-acuity, low-occurrence patient types [38,39,40,41]. 

This scenario lends itself to specialization, and with the HRRT, we were able to implement a number of strategies broadly intended to support skill retention. This included a concentrated exposure to infrequent patient types, performance feedback, and in-situ simulation, all of which facilitate deliberate practice [42,43,44] and have an evidentiary basis in promoting the development of expertise [45,46,47]. 

The use of a safety officer, call–response checklists, principles of crisis resource management, and post-event debriefings also helped to mitigate human factors that have been described in the patient safety literature [21,22] as contributing to adverse events. Collectively, however, these strategies are difficult to use at scale and may not be necessary in the majority of cases where patient acuity is lower. There is, thus, a compelling argument to be made for specialization; however, this comes with certain challenges.

Paramedic service delivery in our context has favored standardization for a number of years through, for example, the use of Lean Six Sigma [48] approaches that aim to optimize efficiency and reduce variation attributable to human factors. Specialization—and, by extension, expertise—viewed through the lens of standardization, is a system fault, and this means that the introduction of a specialized team in our setting contrasted sharply with the existing organizational culture. 

The result was ambiguity around the role of the team and how it fit into the larger system, a problem exacerbated by the speed with which the HRRT was developed. This could have been ameliorated through improved communication and change management. From a practical perspective, transport of the ‘specialists’ to the calls they needed to attend was also challenging. Issues related to staffing, system demands, resource availability, and even traffic meant that ensuring the prompt arrival of an HRRT crew could be difficult and depended heavily on the ability of the team members to ‘hustle’. 

The result was a high-risk AGMP capture rate of slightly under 70%, and a missed case rate that—interestingly—did not vary significantly as a function of team staffing. This suggests that HRRT crews were engaging thoughtfully in processes to triage calls to attend. We did receive feedback, however, that system pressures and a large number of ‘micro-decisions’ in choosing calls to attend were fatiguing for the crews.

Finally, attending calls for patients with life-threatening illness has been cited in several studies as an example of a critical incident [31,49,50,51], exposure to which is associated with an increased risk of post-traumatic stress disorder (PTSD) among paramedics [49,52]. We know, however, that the stress associated with critical incidents and the resulting risk of mental health sequelae is ameliorated by, among other things, mitigating chronic workplace stressors [53], facilitating social support [52,54], and providing unstructured time after critical incidents to decompress with peers [55]. 

With the acknowledged limitation that we did not rigorously investigate the effects of team membership on mental health, the feedback from members was that the team environment was perceived as being helpful. Working in a small group of peers, with a defined mission, a high degree of autonomy, and the opportunity to collaboratively debrief calls appeared to provide members with ways to cognitively reframe what has historically been conceptualized as a potentially traumatic exposure into an opportunity for professional development. This role of teamwork in mitigating critical incident stress is worth exploring further.

## 8. Limitations

Our findings should be interpreted within the context of certain limitations. First, the development and implementation of this program is inherently situated, and care should be taken in extrapolating our processes or findings to other settings. Second, our approach to program evaluation does not support causal inferences. Although we compare procedural success rates as an indicator of quality of care, the observational approach to pre vs. post and captured vs. missed cases makes attributing the differences we observed specifically to the influence of the team difficult. Third, the value of advanced airway maneuvers as an indicator of quality of care is debatable given the broader uncertainty on the role of intubation (for example) on patient outcomes in the pre-hospital setting [36,56].

## Figures and Tables

**Figure 1 ijerph-19-03004-f001:**
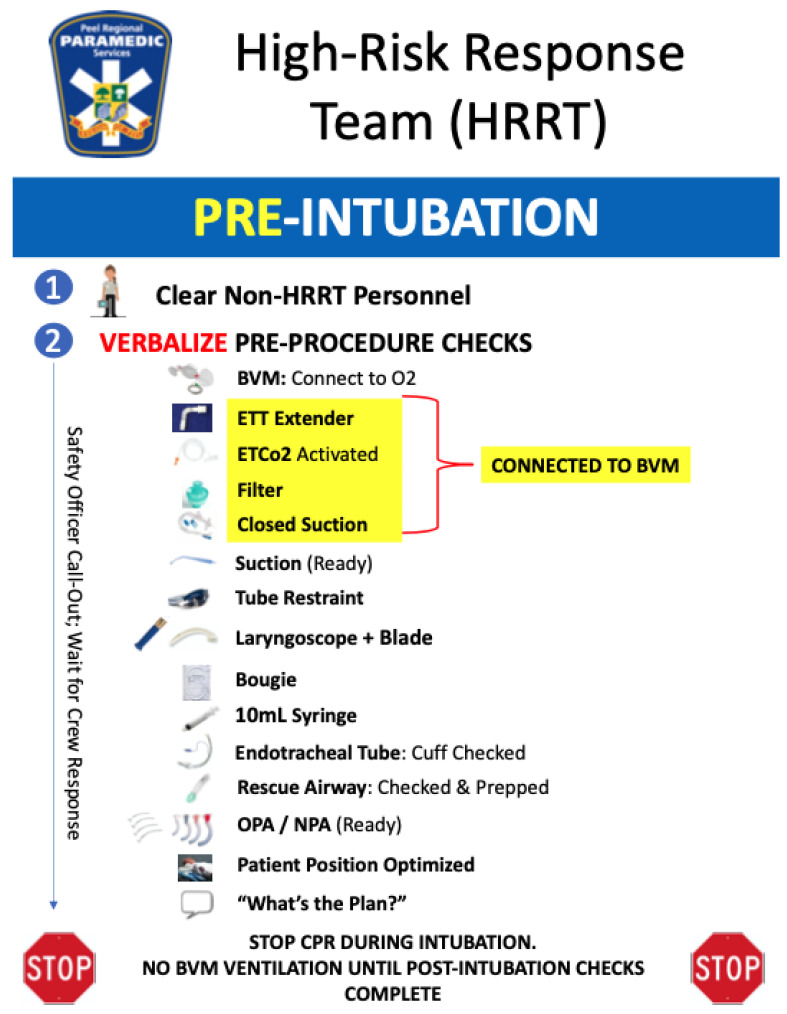
Example of a pre-procedure call–response checklist.

**Figure 2 ijerph-19-03004-f002:**
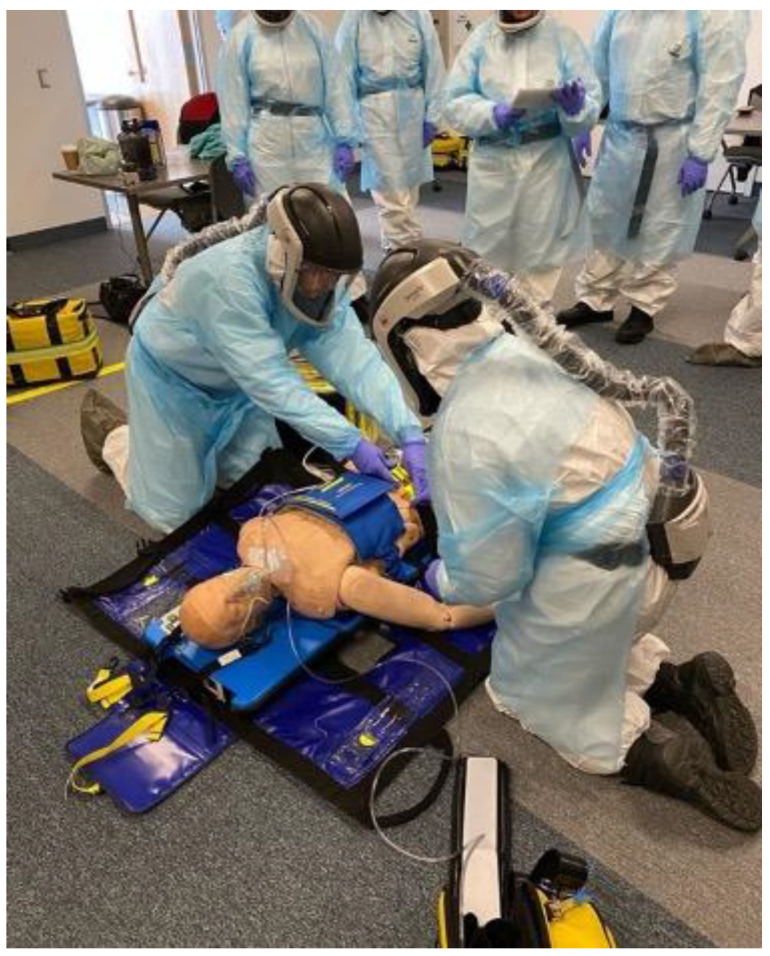
Onboarding training of HRRT personnel. Pictured are two ACPs transitioning from manual to automated CPR with the safety officer monitoring the scene in the background.

**Table 1 ijerph-19-03004-t001:** Recommendations for airway management in patients suspected or confirmed to have COVID-19 distilled from non-systematic narrative literature review. Categorized into feasibility for regular-duty paramedics (Non-HRRT) vs. HRRT within system constraints at the start of the pandemic. * HRRT crews adapted this recommendation by designating ‘hot’ and ‘warm’ zones with controlled access and egress.

Recommendation	Non—HRRT	HRRT
Limit the number of providers during patient care		X
Provide care in negative pressure, airborne isolation rooms with anterooms for donning and doffing PPE		X *
Consider using specialized teams for airway management and have the most experienced clinician perform intubation		X
Avoid bag-mask and non-invasive ventilation		X
Intubate early in the clinical course (particularly during cardiac arrest)	X	X
Use video laryngoscopes for intubation to increase provider distance from the patient		
Use cognitive aids and checklists during high-risk procedures		X
Have a designated safety officer oversee patient care and PPE donning/doffing		X
Conduct regular, in-situ simulation-based team training		X
Use mechanical CPR devices to limit provider exposure to high-risk patients		X
Use airborne PPE with an N95, FFP3, or Powered Air-Purifying Respirator (PAPR)	X	X

**Table 2 ijerph-19-03004-t002:** Aerosol-Generating Medical Procedures defined for the purposes of our program as ‘high-risk’.

‘High-Risk’ AGMPs
Endotracheal Intubation
Supraglottic Airway Insertion
Bag-Valve Mask Ventilation
Continuous Positive Airway Pressure (CPAP)
Open Suction
Cardiopulmonary Resuscitation (CPR)
Tracheostomy Care

## Data Availability

Data is available on request due to legislative requirements within the Province of Ontario and specific regulations in Region of Peel that govern data security and access. The data presented in this study are available on request from the corresponding author. Note this may require a written data sharing agreement with the Region of Peel’s legal department.

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
