# Peer review of "Protected 911: Development, Implementation, and Evaluation of a Prehospital COVID-19 High-Risk Response Team"

_ijerph, 2022, doi:10.3390/ijerph19053004_

Round 1
Reviewer 1 Report
Please find my peer review comments attached.

Author Response
Dear colleague,
Thank you for taking the time to review our paper. Please see the attachment for a detailed response to your comments.
Many thanks,
Justin

Reviewer 2 Report
I read the article with great interest, I believe that the experience is well presented and useful for an international audience and therefore should be published.
In the last two years clinical risk management and attention to risk mitigation strategies has increased and made necessary and central in the activities, the authors could cite doi: 10.3390/healthcare9010017 this work that explains the efforts made in many countries to increase security in a pandemic approach.
I believe that the authors could improve the text by adding some indication if possible on the costs incurred for this type of equipment and if this type of situation can be exported to other hospital contexts also in other parts of the world and with what benefits
Author Response
Dear colleague,
Thank you for reviewing our paper. Please see the attachment for a detailed response to your comments.
Many thanks,
Justin

Reviewer 3 Report
The article contains a description of the case study, however, it requires very significant improvement. First, the overall purpose of the whole analysis is not clear:
- what is to be proved?
- how is science expected to be contributed?
- to what extent does the presented analysis permanently expand the available knowledge?
The described organizational solution must be based on theoretical ground (with references to the literature):
- why were these (and not other) specific goals selected for the implementation of the project? (so far there is no sufficient theoretical justification based on the literature in the text)
- why were these specific solutions chosen (and which ones were dropped and why)?
- what was the process of reaching the chosen solution and who participated in this process?
- Were any methods used in this process, e.g. brainstorming etc.?
The scope of the analyzed data raises reservations: a comparison of the period before and after the project implementation may lead to wrong conclusions. The initial weeks of gathering experience during the pandemic were associated with very dynamic changes in recommendations, beliefs, etc. A better solution is to compare the effects of HRRT type work with the effects of work of another (traditional) crew, assuming that the observation took place during the same period of time (where the traditional crew could operate in a different but similar terrain).
The phenomenon of a change in the level of motivation and efficiency of a team's performance as a result of an invitation to collaborate in an experiment has been widely described in the literature since the Hawthorne experiment in 1924.
Author Response

(The authors gave the same response as above.)

Round 2
Reviewer 3 Report
Thank you for clearing up my doubts. I propose to consider eliminating these doubts in the article itself even more broadly and clearly than after the last modification. In order to demonstrate the greater scientific importance of the article, I suggest that in the conclusions section, the following thesis (or a similar one) should be used: "the analysis of the results of the conducted research allows to confirm the Hawthorne effect also in the case of a highly specialized professional group of paramedics working in conditions of risk of losing health and life". As for editorial notes, it is necessary to organize the reference numbering (especially after the last text changes).
Author Response
Dear colleague,
Thank you again for your helpful feedback. We have added the language you recommended to the conclusion section of the manuscript.
With thanks,
Justin